# New Challenges for Classical and Quantum Probability

**DOI:** 10.3390/e24101502

**Published:** 2022-10-21

**Authors:** Luigi Accardi

**Affiliations:** Centro Vito Volterra, University Roma Tor Vergata, 00133 Roma, Italy; accardi@volterra.uniroma2.it

**Keywords:** quantum probability, orthogonal polynomials, quantum decomposition of a classical random variable

## Abstract

The discovery that any classical random variable with all moments gives rise to a full quantum theory (that in the Gaussian and Poisson cases coincides with the usual one) implies that a quantum–type formalism will enter into practically all applications of classical probability and statistics. The new challenge consists in finding the classical interpretation, for different types of classical contexts, of typical quantum notions such as entanglement, normal order, equilibrium states, etc. As an example, every classical symmetric random variable has a canonically associated conjugate momentum. In usual quantum mechanics (associated with Gaussian or Poisson classical random variables), the interpretation of the momentum operator was already clear to Heisenberg. How should we interpret the conjugate momentum operator associated with classical random variables outside the Gauss–Poisson class? The Introduction is intended to place in historical perspective the recent developments that are the main object of the present exposition.

## 1. Introduction: Quantum Theory and Non-Kolmogorov Probabilities

More than 40 years have passed since the first published papers on the quantum probabilistic approach to the foundational problems of quantum mechanics, and the path leading to the present developments has been long and tortuous. Therefore a short synthesis of the salient moments of this path may be not useless, especially for younger generations.

### 1.1. Mathematical Models: Of Space and of the Laws of Chance

Until the early 1960s, the preferred tool for illustrating the (apparent) paradoxes of quantum mechanics was the two-slit experiment, first used by Einstein in the 1927 Solvay conference with an emphasis on non-locality, then elaborated by Heisenberg in his book Physics and Philosophy with emphasis on virtual reality, and further by Feynman [1], who arrived at the same conclusion as Heisenberg through a contradiction between the experimental data and classical probability.

In the mid-1960s, Bell’s inequalities [2] became very popular because it was believed that they could provide an experimental proof of the impossibility of local hidden variable theories, explaining EPR-type correlations.

The first step of the quantum probability approach (QP in the following) to the foundational problems of quantum theory consisted in the understanding that both these apparent paradoxes had the same root and that this root had nothing to do with virtual reality or non-locality, but was rather of probabilistic nature: in both cases, a probabilistic inequality was violated, and this was considered to be paradoxical because the deduction of this inequality was believed to be based on obvious assumptions (i.e., that at each time, an electron has a position or that a spin in a given direction has a precise value) and on straightforward classical probabilistic computations.

The first public exposition of this approach, with distribution of the corresponding preprint [3], took place in 1981 at the Salamanca Workshop Foundations of Quantum Mechanics, attended, among others, by Aspect, Bell, Hiley, Piron, …. The main observation was very simple, namely, that in both cases:–the experimentally evaluated probabilities came from three mutually incompatible physical situations (choices of open slits or of the polarization directions);–the probabilistic calculation implicitly postulated the existence of joint probabilities (equivalently classical probabilistic model) for the events involved.

The simple remark was that the existence of joint probabilities implies the existence of mathematical constraints among the measured statistical parameters (e.g., conditional probabilities or correlations). Therefore, to postulate the existence of such joint probabilities is an unwarranted assumption.

In other words, the contradictions discovered by the founding fathers of QM were not between experimental data and the explicit assumption of reality (or locality) but between these data and the implicit assumptions that one can apply the rules of classical probability to these data, or equivalently, that all these experimental data can be fitted into a single Kolmogorov probabilistic mode, which requires a precise mathematical description that depends on the type of experimental data considered, e.g., conditional probabilities, correlations, etc.

To sum up, both the classical and the quantum probabilistic model are perfectly coherent in themselves (as much as a mathematical model can be). Apparent paradoxes arise when they are mixed without the necessary precautions. What are these precautions?

The answer provided to this question by QP was inspired by what happened slightly more than 100 years before in geometry, with Gauss’ discovery of the first geometrical invariant: curvature. This analogy, strongly emphasized since the very early 1980s, led to the idea of statistical invariants, i.e., necessary and sufficient conditions for a set of experimental data to be describable within a single Kolmogorov model (informally, to admit joint probabilities). Except in special cases, these are as difficult to calculate in concrete cases as are geometrical invariants, however, in both cases there are now several concrete examples.

Although the mathematical frameworks in geometry and probability are quite different, the conceptual message is the same: the available experimental data impose constraints on the mathematical models one can use to describe the physical situation which produced these data. The specific nature of these data is not relevant; they are typically angles (or distances, etc.) in geometry and conditional probabilities (or correlations, etc.) in probability. In both cases, the invariants are functions of the experimental data, and different mathematical models are classified according to the ranges of these functions.

The first examples of statistical invariants are contained in [3], and it is a surprising fact that they are related to geometrical invariants. Perhaps this is a manifestation of a more general phenomenon.

**Remark** **1.**
*The fact that there are constraints among classical joint probabilities was already clear in the first treatises on probabilities, such as Cardano’s monograph from 1400. The precise mathematical formulation of these constraints has undergone a long elaboration, involving, among others Pascal, Fermat, Boole, Bayes, etc., and culminating in the Kolmogorov consistency conditions, which deal with joint probabilities. Several years after the first articles on quantum probability appeared, authors have embroidered over this fact in hindsight, producing articles that can be fun for those interested in a posteriori reconstructions of history.*


It is, however, a historically documented fact that these consistency conditions were first applied to the analysis of the apparent quantum paradoxes in the framework of quantum probability, and that nowadays it is beginning to be generally accepted that this hybridization has produced, and is still producing, fructuous developments in both quantum theory and probability.

From Kolmogorov consistency conditions we can deduce other conditions that do not involve joint probabilities, and instead involve other types of statistical data. For example, these kinds of conditions were considered in statistics for correlations and in [3] for conditional probabilities. Thus, many results in the classical case can be interpreted as calculations of statistical invariants. What changes with quantum probability is the perspective: in classical probability, these results are coherence conditions among the experimental data; in quantum probability, they are conditions for the existence of a given mathematical model permitting description the experimental data. The idea that such compatibility conditions must be looked at for the quantum probabilistic model as well was first advanced in [3], and the corresponding invariant was computed for experimental data provided by transition probabilities. In [4], it was shown how to proceed, in the EPR case, from transition probabilities to correlation functions (and conversely). A few years later, Tsirelson and Khalfin proved a similar constraint in the quantum model for experimental data provided by the EPR correlations [5], showing that the bound in Bell’s inequalities is the best possible one for that type of correlations.

### 1.2. Hidden Axioms in Classical Probability

Before quantum probability, several mathematical models were proposed to extend the usual Hilbert space model of quantum theory in different directions: non-commutative projective geometries, extensions of Boolean algebras (quantum logics), partially ordered sets, von Neumann algebras, etc.

One of the main tenets introduced by QP is to strictly distinguish between:–description of a mathematical model;–statements of model-independent axioms (from which the possible mathematical models are deduced).

For example, Kolmogorov’s 1933 monograph [6] answers, for classical probability, the former of the above questions, though not the latter.

The idea of the QP program was to realize for probability what Hilbert had realized for geometry almost one century before,. This in fact was the sixth of the 23 problems formulated by Hilbert in the 1900 Paris International Congress of Mathematicians [7]: … to treat axiomatically those physical disciplines in which already today mathematics plays a predominant role … these are in the first place the calculus of probability and mechanics … (see [8] for a discussion of this issue).

From the early years of QM it was clear that its mathematical model was describing a new probability calculus that was quite different from the classical one (see, e.g., [9]). Then, if we believe along with Hilbert that model-independent axioms should determine the mathematical model, we are forced to conclude that among the (model-independent) axioms of classical probability are those that are violated in the quantum mechanical model, and from this naturally rises the following question: … which axiom plays for probability the role played by the parallel axiom in geometry?

The QP answer to this question is the Bayesian definition of conditional probability.

In fact, in classical probability Bayes’ formula is not considered as an axiom, but as the definition of conditional probability [6]. In [10], it is shown that to deduce Bayes’ formula one needs five model-independent axioms, and for each of these a counter-example based on the quantum probabilistic formalism is deduced.

### 1.3. Model-Independent Axioms for Quantum Probability

The conclusion in the previous section is a negative one: Bayes’ formula does not hold in quantum probability. The natural problem arising from this finding is, which is a minimal set of model-independent axioms from which the standard mathematical model of quantum mechanics can be deduced? This problem was first solved in [11] and subsequently extended, improved, and commented on in several papers (see the recent exposition [8] and the bibliography therein). In these papers:(i)A set of simple model-independent and physically meaningful axioms that unify classical and quantum probability is proposed.(ii)A classification theorem is proved that exhibits all models for these axioms.(iii)Although it is shown that the deduced models include the known models of classical and quantum probability, interesting new possibilities emerge.

The new axiomatization extends Schwinger’s proposal to use the notion of measurement rather than of event to describe quantum mechanics to the whole theory of probability. In it, Schwinger’s qualitative idea is translated into mathematical axioms which allow for the deduction of a set of equations with solutions that classify the probability models described by the given axioms. The main new results with respect to Schwinger’s paper are the formulation of the axioms, the deduction from them of the structure equations, and most importantly, their solutions which classify the possible probabilistic models that satisfy a weak form of Heisenberg’s indeterminacy principle.

### 1.4. The Physical Roots of the New Probabilistic Axioms: Statistics of Adaptive Systems

The hardest challenge to the QP approach, still open in the late 1980s and not surprisingly the last to be solved, is again better explained exploiting the analogy between the development of geometry and that of probability.

After the discoveries of non-Euclidean geometries, and especially after Gauss’ Theorema Egregium, it became clear that the main distinctive feature between Euclidean and non-Euclidean geometries is curvature. However a clear idea of which physical agent might give rise to a curved space was missing until general relativity filled this gap by relating curvature to strong concentrations of matter. Curvature is the main geometrical invariant measuring how much the parallel axiom is violated. As we know that in probability Bayes’ formula plays the role of the parallel axiom in geometry, the natural thing to do is to look for statistical conditions that justify violations of the Bayes formula.

The discovery of such conditions comes from the understanding that the statistics of passive properties can be quite different from the statistics of the responses, and that the latter can easily violate the classical probabilistic predictions. The chameleon metaphor was created to intuitively explain the differences between passive and adaptive dynamic systems.

To get a feeling for this, suppose that a box contains a large number of chameleons and that two experimentalists *A* and *B* know that these animals can be brown or green, but do not know their adaptive properties and use for them the same categories used for the colors of balls (i.e., a passive property). They want to experimentally estimate the fraction of brown and green chameleons in the box, however, in their experiments they can only lure one chameleon out of the box at a time and can do this only by putting food either on a (green) leaf or on a (brown) piece of wood. It is clear that the results of their measurements depend on which measurements they decide to perform. For example, if *A* decides to always use the leaf and *B* to always use the piece of wood, *A* will conclude that the chameleons are green with probability 1 and *B* that they are brown with the same probability. Only if they agree a priori to use the leaf in a fraction p∈[0,1] of their measurements and the piece of wood in the remaining fraction, and if they both know which color is measured in each experiment, will they find non-contradictory results. However, if their local measurements are independent, then they will find mutually contradictory results with probability 1.

The example above describes a simple but extreme situation, and provides a good intuitive idea of the main thesis of quantum probability: the statistics of responses can violate classical probability rules, thus leading to contradictions if you try to apply them, while the statistics of passive properties, symbolized by the urns containing different mixtures of green and white balls, cannot. Moreover, this example can be simply modified to include more complex situations in which randomness in the results of measurements is involved.

The non-trivial mathematical problem was whether classical adaptive dynamical systems can reproduce the EPR correlations while fully respecting locality and causality?

The positive answer to this question is in the paper [12] (see [13] for further comments). With these papers, the cycle of answers to the apparent paradoxes that plagued QM for little more than half a century can be considered closed.

Those who feel psychologically attracted by notions such as virtual reality, non–locality, etc., continue to play with these notions. The only difference is that now theorems and experiments unequivocally prove that the claim that these beliefs are supported by experimental data is unwarranted.

### 1.5. Non-Kolmogorovianity Outside Quantum Physics

The problems raised by the founding fathers of Quantum Theory have played a fundamental role in our understanding of the connections between QM and probability theory. The understanding that the laws of chance can have different mathematical models and that different models correspond to different types of statistics (of passive or response type) is as profound as the analogue understanding that took place for the laws of space about two centuries ago.

As is usual in science, the solution of an age-old problem opens new problems, certain of which are as deep and challenging as the problems of the foundations of quantum mechanics were at the beginning of the 20th century. There is no doubt that one of the main challenges among these will be the search for applications of non-Kolmogorovian statistics outside the quantum domain. This program was first enunciated as an abstract possibility in the International Conference Quantum Probability and Noncommutative Geometry, (Official satellite meeting of International Mathematical Physics Congress), August 2000, Nottingham Trent University, and later in a more systematic way in [14]. The merit of having transformed this abstract possibility into a concrete line of scientific research goes entirely to Andrei Khrennikov, who found the first candidate for such applications (see [15,16]) and later discovered many other ones in the fields of economics, psychology, sociology, and different sectors of biology, (see [17,18,19,20,21,22] and the more recent publications [23,24,25,26]), thus giving birth to a new scientific research direction that is now flourishing and will surely attract more and more people for whom understanding is more important than formalism.

Up until now, the only sufficiently developed non-Kolmogorov models were the classical and the quantum one. However, now that we know that the quantum model is only one among many other possibilities (which, as shown in the second part of this paper, are as natural as the quantum ones), the problem is raised of investigate whether the classical and quantum models are the only candidates to account for the specific features of the kind of response statistics naturally emerging, for example, in molecular biology. The fact that the notion of maximally observable, crucial in both the classical and the quantum model, is not easy to define in a biological context, may lead to doubts about this and suggest the possibility that a new statistics might meet the needs of this discipline better than classical or quantum ones. This is surely a problem to be kept in mind.

## 2. Deeper Levels of Classical Probability

The goal of the first section of this paper has been to provide a quick glance at the process which, starting from the attempt to explain the apparent quantum paradoxes, gradually led to the emergence of non-Kolmogorov probabilities, to the statistical invariants as a tool to discriminate among different statistical models, to a unified axiomatization of classical and quantum probability, and to the first concrete examples of adaptive and deterministic classical dynamical systems that reproduce the EPR correlations.

However, the fact that the statistical invariants of the classical and quantum probabilistic models are different does not exclude the possibility that one mathematical model can be embedded into the other while preserving all statistical properties (isomorphic stochastic embedding). For example, it is known that two classical random variables $X,X^{\prime }$ defined on two probability spaces (Ω,F,P), $\left(\Omega^{\prime },F^{\prime },P^{\prime }\right)$ are identified if they have the same distribution (stochastic equivalence), i.e., if for each bounded function $f\in L_{C}^{\infty }\left(S,P\right)$ one has
(1)$$\int_{\Omega }f\left(X\right)dP=\int_{\Omega^{\prime }}f\left(X^{\prime }\right)dP^{\prime }$$

Thus, for any real-valued random variable *X* with probability distribution $P_{X}\in \mathrm{Prob}\left(R\right)$, one has
(2)$$\int_{\Omega }f\left(X\right)dP=\int_{R}f\left(Q\right)dP_{X}=\langle \Phi ,f\left(Q\right)\Phi \rangle$$
where Φ is the constant function equal to 1 and *Q* is the multiplication operator (Qf)(x):=xf(x) ($f\in L_{C}^{2}\left(S,P\right)$). As the mapping $f\in L_{C}^{\infty }\left(S,P\right)\mapsto f\left(X\right)\in B(L_{C}^{2}\left(C,P_{X}\right)$ (with bounded operators on $L_{C}^{2}\left(C,P_{X}\right)$ is an injective ∗–homomorphism, the identity Equation (Equation 2) can be considered as an embedding of the classical probability of a single real-valued random variable into the quantum probabilistic model. The above argument can be easily adapted to an arbitrary classical stochastic process, providing a natural embedding of the whole classical probabilistic model into the quantum one.

All this has been well known from the early developments of the mathematical formalism of quantum theory, and is at the basis of the folklore statement that the quantum probabilistic model is an extension of the classical one.

What is quite non-trivial and unexpected is that the converse of this statement is true, namely, the fact that the standard quantum formalism arises as very special case of a deeper level of the classical probabilistic model.

The second part of this paper is devoted to precisely formulating this statement and to illustrating why it holds in the simplest possible context of quantum systems with 1 degree of freedom.

### New Developments: Statement of the Problem

The two building blocks of Quantum Mechanics are Heisenberg commutation relations and the Schrödinger equation. The original Heisenberg formulation of commutation relations was in terms of position and momentum, namely,
(3)[p,q]=pq−qp=−iℏ(1-degreeoffreedom)
where ℏ>0 is a universal constant, called Planck’s constant (divided by 2π). The fact that ℏ≠0 implies that *p* and *q* do not commute. In this sense, we can say that Heisenberg commutation relations have opened the way to non-commutativity in physics. The question of where do they come from? accompanied quantum theory from its very beginning. The fact that the founding fathers of quantum mechanics were not able to answer this, or related questions such as, from where does non-commutativity in QM arise? why does it take the very special form given by Equation (Equation 3)? created a sense of dissatisfaction that has lasted for a long time. Evidence of this dissatisfaction can be found in many of Heisenberg’s writings; for example, in 1927 he wrote: ”The mathematical formalism of quantum mechanics is far from intuition …”. This aura of mystery around the origins of the commutation relations Equation (Equation 3), and hence of the whole mathematical formalism of quantum theory, persisted for more than 50 years, as testified by Nelson’s known aphorism (1970): 2D quantization is a functor, (he was thinking of the Fock functor) 1st Quantization is a mystery.

In the present paper, we will see that:(1)There is no more mystery.(2)The commutation relations discovered by Heisenberg are a very special case of a universal phenomenon of classical probability theory.(3)This new fact opens fascinating new challenges for all fields where classical probability plays a role, in particular economics, sociology, psychology, machine learning, artificial intelligence, image reconstruction, etc.

For simplicity of exposition, I only discuss the case of a single real valued random variable. In the language of physics, this corresponds to systems with 1 degree of freedom.

All the results continue to hold for systems with arbitrarily many degrees of freedom (see [27] and the bibliography therein), however, in this paper we will not discuss this extension. The landscape in this case, i.e., for random variables with values in a vector space of dimensions greater or equal 2 (in particular, infinity, so that quantum fields are included), becomes much more interesting. In particular, aside from the usual commutation relations (arising from the fact that commutators between creators and annihilators are gradation-preserving), a new class of commutation relations arises. The reason why this class did not show up in usual quantum mechanics is that usual quantum theory coincides with the quantum theory canonically associated with Gaussian fields (see Section 3.6 below). This implies that usual quantum theory is included in the strictly larger family of quantum theories canonically associated with measures (such as Gaussians) that, with an affine transformation, can be transformed into product probability measures, and for this class of theories the new commutation relations are automatically satisfied.

## 3. The Main Results

### 3.1. Relevant Notations in Classical Probability

In the following, we fix: a classical real valued random variable *X*; the probability distribution of *X*, denoted μ∈Prob(R); we assume that X≡μ has moments of all orders, i.e.,
$$E({\left|X\right|}^{n}){=\int |x|}^{n}\mu \left(dx\right)\equiv {\mu (|X|}^{n})<+\infty$$

Here and in the following, we denote with the same symbol the probability measure μ and the associated integral. Define
$$P\equiv P_{X}:=\;\left\{*\mathrm{-algebra}\;\mathrm{of}\;\mathrm{complex}\;\mathrm{valued}\;\mathrm{polynomial}\;\mathrm{functions}\;\mathrm{of}\;X\right\}$$(for the point-wise arithmetic operations and complex conjugation).

Here and in what follows, we use the symbol ≡ to mean that two notations are used indifferently. Because *X* has moments of all orders, the map
$$P\in P\mapsto \mu \left(P\right):=\int_{R}P\left(x\right)\mu \left(dx\right)\;\in C$$
is a state on P, i.e., a positive linear functional μ:P→C such that $\mu (1_{P})=1$. The state μ induces on P the semi-scalar product
(4)$$\langle P,Q\rangle :=\mu \left(P^{*}Q\right)=\int_{R}\bar{Q(x)}P\left(x\right)\mu \left(dx\right)$$
and hence, a structure of semi-Hilbert space on P (possible zero-norm elements). From now on, for simplicity, we only consider the case of a scalar product.

### 3.2. Orthogonal Polynomials

Using the scalar product Equation (Equation 4) to orthogonalize the monomials $\left(X^{n}\right)$ without normalization, one obtains a sequence of polynomials
$$\Phi_{n}:=X^{n}-P_{n-1]}\left(X^{n}\right)$$
where $P_{n-1]}\left(X^{n}\right)$ denotes the component of $X^{n}$ on the sub-space of all polynomials of degree less than or equal to n−1.

The sequence $(\Phi_{n})\subset P$ is called the sequence of monic μ-orthogonal polynomials.

### 3.3. Jacobi Monic 3-Diagonal Relations

Carl Gustav Jacob Jacobi proved the following Theorem 150 years ago (∼1870).

**Theorem** **1.**
*For any probability measure on R with all moments, the following monic 3-diagonal relations hold:*

(5)
$$x\Phi_{n}\left(x\right)=\Phi_{n+1}\left(x\right)+\alpha_{n}\Phi_{n}\left(x\right)+\omega_{n}\Phi_{n-1}\left(x\right)\;,\;x\in R$$

*where*

(6)
$$(\alpha_{n})\subset R\;(\mathrm{called}\;a\;\sec\mathrm{ondary}\;\mathrm{monic}\;\mathrm{Jacobi}\;\mathrm{sequence})\subset R$$

*is an arbitrary real sequence and*

$$\left(\omega_{n}\right)\subset R_{+}\;(\mathrm{called}\;a\;\mathrm{principal}\;\mathrm{monic}\;\mathrm{Jacobi}\;\mathrm{sequence})$$

*is a sequence of real numbers satisfying*

(7)
$$\omega_{n}\geq 0\;;\;\omega_{n}=0\;\Rightarrow \;\omega_{k}=0\;,\;\forall k\geq n$$



**Remark** **2.**
*The term monic means that $\Phi_{n}\left(x\right)$ is a polynomial of degree n and the leading coefficient of $x^{n}$ is 1.*


About 70 years after Jacobi (∼1935), Favard proved the converse of Theorem (1).

**Theorem** **2.**
*(Favard) Given two sequences $\left(\omega_{n}\right)$, $\left(\alpha_{n}\right)$ satisfying Equations (Equation 6) and (Equation 7), there exists a probability measure μ on R with orthogonal polynomials satisfying the monic 3-diagonal relations Equation (Equation 5) with the given two sequences $\left(\omega_{n}\right)$, $\left(\alpha_{n}\right)$.*


**Remark** **3.**
*In the case of monic Hermite polynomials (the orthogonal polynomials of the Gaussian measure), all $\alpha_{n}$ are zero and $\omega_{n}=\sigma n$, with σ>0 (see the proof of Theorem 4 below).*


Conclusion: (good) probability measures on R are parameterized by pairs of sequences $\left(\omega_{n}\right)$, $\left(\alpha_{n}\right)$. In less favorable cases, more than one probability measure on R can be associated with the same pair of sequences.

### 3.4. The Canonical Quantum Decomposition of a Classical Random Variable

The Jacobi identity
$$x\Phi_{n}\left(x\right)=\Phi_{n+1}\left(x\right)+\alpha_{n}\Phi_{n}\left(x\right)+\omega_{n}\Phi_{n-1}\left(x\right)$$
naturally suggests the introduction of the following operators acting on the pre-Hilbert space $\left(P,{\langle \;\cdot \;,\;\cdot \;\rangle }_{\mu }\right)$ (a space with a scalar product, though not necessarily complete with respect to it):$$\left(X\Phi_{n}\right)\left(x\right):=x\Phi_{n}\left(x\right)\;\mathrm{Multiplication}\;\mathrm{operator}\;\mathrm{by}\;X$$
$$\left(a^{+}\Phi_{n}\right)\left(x\right):=\Phi_{n+1}\left(x\right)\;\mathrm{creation}\;\mathrm{operator}$$
$$\left(a^{0}\Phi_{n}\right)\left(x\right):=\alpha_{n}\Phi_{n}\left(x\right)\;\mathrm{preservation}\;\mathrm{operator}$$
$$\left(a^{-}\Phi_{n}\right)\left(x\right):=\omega_{n}\Phi_{n-1}\left(x\right)\;\mathrm{annihilation}\;\mathrm{operator}$$

Here, we use the notation
$$\{a^{+},a^{-},a^{0}\}:=\;\mathrm{the}\;\mathrm{CAP}\;\mathrm{operators}\;(\mathrm{Creation},\;\mathrm{Annihilation},\;\mathrm{Preservation})$$

In terms of CAP operators, the Jacobi relations become
(8)$$X\Phi_{n}=a^{+}\Phi_{n}+a^{0}\Phi_{n}+a^{-}\Phi_{n}\;,\;\forall n\in N$$

Because $\left(\Phi_{n}\right)$ is an ONB, the vector identity Equation (Equation 8) is equivalent to the operator identity
(9)$$X=a^{+}+a^{0}+a^{-}$$

**Definition** **1.**
*The identity Equation (Equation 9) is called the canonical quantum decomposition of the classical random variable X.*


**Remark** **4.**
*A classical random variable X usually admits many quantum decompositions, however, the canonical one is unique up to the natural definition of isomorphism of two quantum decompositions (not discussed in this paper; see [28]).*


I include the simple proof of the following theorem because it plays a key role in the whole development of the theory.

**Theorem** **3.**
*The adjoint of $a^{+}$ exists and is equal to a. Moreover,*

(10)
$${\left(a^{0}\right)}^{*}=a^{0}$$


(11)
$$a\Phi_{0}=0\;(\mathrm{Fock}\;\mathrm{property})\;$$



**Proof.** The proof of Equation (Equation 10) is clear because, by definition, $a^{0}$ has real eigenvalues $\alpha_{n}$ corresponding to the eigenvectors $\Phi_{n}$, which are a complete orthogonal system. The Fock property Equation (Equation 11) follows from the first statement and the fact that the $\left(\Phi_{n}\right)$ are an orthogonal basis. In fact, for each n∈N
$$\langle \Phi_{n},a\Phi_{0}\rangle =\langle a^{+}\Phi_{n},\Phi_{0}\rangle =\langle \Phi_{n+1},\Phi_{0}\rangle =0$$It now remains to prove the first statement, i.e., that ${\left(a^{+}\right)}^{*}=a$. Using Equation (Equation 10), this follows from
$$X=a^{+}+a^{0}+a=X^{*}={\left(a^{+}\right)}^{*}+a^{0}+a^{*}\Leftrightarrow a^{+}-{\left(a^{+}\right)}^{*}=a^{*}-a$$However, from the definition of adjoint, for each n∈N, $\left(a^{+}-{\left(a^{+}\right)}^{*}\right)\Phi_{n}\in C\cdot \Phi_{n+1}$ and $\left(a^{*}-a\right)\Phi_{n}\in C\cdot \Phi_{n-1}$. Therefore, orthogonality implies that the identity $\left(a^{+}-{\left(a^{+}\right)}^{*}\right)\Phi_{n}=\left(a^{*}-a\right)\Phi_{n}$ can hold iff $a^{+}={\left(a^{+}\right)}^{*}$ and $a^{*}=a$. □

The quantum decomposition of the classical random variable *X*
(12)$$X=a^{+}+a^{0}+a^{-}$$
means that every classical random variable has a microscopic structure as a sum of three (CAP) operators.

We will see that these CAP operators do not mutually commute.

Conclusion: non-commutativity arises naturally from classical probability.

Furthermore, much more than generic non-commutativity holds. Let us prove how, from classical probability, one can deduce a very explicit multiplication table for these operators; this is much more than a set of commutation relations.

### 3.5. Commutation Relations Canonically Associated with the Classical Random Variable X

Let us start from the quantum decomposition of the classical random variable
$$X=a^{+}+a^{0}+a^{-}$$
$$\Leftrightarrow X\Phi_{n}=\Phi_{n+1}+\alpha_{n}\Phi_{n}+\omega_{n}\Phi_{n-1}\;,\;\forall n\in N$$
and recall the definition of the monic CAP operators:$$a^{+}\Phi_{n}=\Phi_{n+1}\;;\;a^{-}\Phi_{n}=\omega_{n}\Phi_{n-1}\;;\;a^{-}\Phi_{0}=0$$

This implies the multiplication table
(13)$$a^{-}a^{+}\Phi_{n}=a^{-}\Phi_{n+1}=\omega_{n+1}\Phi_{n}$$
(14)$$a^{+}a^{-}\Phi_{n}=\omega_{n}a^{+}\Phi_{n-1}=\omega_{n}\Phi_{n}$$
with the convention, from now on standard:$$\omega_{0}:=0$$

Subtracting Equation (Equation 14) from Equation (Equation 14), one finds the ω–commutation relations
$$\left(a^{-}a^{+}-a^{+}a^{-}\right)\Phi_{n}=\left(\omega_{n+1}-\omega_{n}\right)\Phi_{n}$$
(15)$$\Leftrightarrow \left[a^{-},a^{+}\right]\Phi_{n}=\left(\omega_{n+1}-\omega_{n}\right)\Phi_{n}$$

As we did with the quantum decomposition, we want to write equation Equation (Equation 15) as an operator identity. To this end, we introduce the number operator
(16)$$\Lambda \Phi_{n}:=n\Phi_{n}\Leftrightarrow F_{\Lambda }\Phi_{n}:=F\left(n\right)\Phi_{n}$$
for any function F:N→C. With this notation, the operator version of the vector ω–commutation relations Equation (Equation 15) becomes
(17)$$\left[a^{-},a^{+}\right]\overset{\left(16\right)}{=}\omega_{\Lambda +1}-\omega_{\Lambda }=:\partial \omega_{\Lambda }$$

Notice that these are operator-valued commutation relations (CR); the commutator $\left[a^{-},a^{+}\right]$ is equal to the operator $\partial \omega_{\Lambda }$.

Conclusion: any classical real valued random variable *X* has its own commutation relations provided by Equation (Equation 17). This naturally leads to the following question:

What about the original Heisenberg commutation relations?

### 3.6. The Heisenberg Commutation Relations

**Theorem** **4.**
*For every strictly positive number ℏ there exists a unique symmetric classical real-valued random variable X for which the associated canonical commutation relations are:*

(18)
$$[a^{-},a^{+}]=\hbar$$


*This is the Gaussian random variable with variance ℏ. (Symmetric means that all odd moments vanish).*


**Proof.** 

$$Equation\;\left(18\right)\Leftrightarrow \forall n\in N\;:\;\hbar \Phi_{n}=\left[a^{-},a^{+}\right]\Phi_{n}=\left(\omega_{n+1}-\omega_{n}\right)\Phi_{n}$$


$$\Leftrightarrow \hbar =\omega_{n+1}-\omega_{n}\Leftrightarrow \omega_{n+1}=\omega_{n}+\hbar$$


$$\Leftrightarrow \omega_{n+1}=2\hbar +\omega_{n-1}=3\hbar +2\omega_{n-2}=\cdots =\left(n+1\right)\hbar$$


(19)
$$\Leftrightarrow \omega_{n}=n\hbar \;\forall n\in N$$



Now, one can use the following result of the theory of orthogonal polynomials. The centered Gaussian random variable with variance *ℏ* is the unique symmetric real-valued random variable with the principal Jacobi sequence
(20)$$\omega_{Gauss,n}=\hbar n\Leftrightarrow \omega_{Gauss,n+1}-\omega_{Gauss,n}=\hbar \;;\;n\in N$$

Combining this statement with Equation (Equation 19), one can conclude that the Gaussian random variable with variance *ℏ* is the unique symmetric random variable with CAP operators that satisfy
$$[a^{-},a^{+}]=\hbar \cdot 1$$

□

Theorem (4) explains why, within the wider perspective of probabilistic quantization, usual quantum mechanics should be called *Gaussian quantum mechanics*.

### 3.7. Momentum Operator Associated with a Classical Random Variable X

**Definition** **2.**
*Let $X=a^{+}+a^{-}$ be a symmetric real valued classical random variable. The symmetric operator*

(21)
$$p_{X}:=i\left(a^{+}-a\right)$$

*is called the **momentum operator** associated with X.*


**Remark** **5.**
*In usual quantum mechanics:*
–
*the multiplication operator from $X=a+a^{+}$ is called the position (or field) operator;*
–
*the operator $P_{X}:=i\left(a^{+}-a\right)$ is called the momentum operator;*
–
*physical observables are in one-to-one correspondence with Hermitean operators (which is why we require Hermiteanity).*


*Definition 2 is justified by the following result.*


**Theorem** **5.**
*For a symmetric classical random variable, one has*

$$\left[X,P_{X}\right]=2i\partial \omega_{\Lambda }=2i\left(\omega_{\Lambda +1}-\omega_{\Lambda }\right)$$



**Corollary** **1.**
*In the Gaussian case ($\partial \omega_{\Lambda }=\hbar >0$), we find*

$$\left[X,P_{X}\right]=2i\partial \omega_{\Lambda }=2i\hbar$$

*(which, up to normalization factor 2, is Heisenberg’s position–momentum commutation relation).*


**Corollary** **2.***X and $P_{X}$ satisfy the uncertainty principle for classical probability, namely, that for all unit vectors*ψ∈Dom(p)∩Dom(q),
(22)$$\left|\langle \psi ,\left[p,q\right]\psi \rangle |=\right|\langle \psi ,\partial \omega_{\Lambda }\psi \rangle \left|=\right|E_{\psi }\left(\partial \omega_{\Lambda }\right)|\leq 4{\mathrm{Disp}}_{\psi }\left(p\right)\cdot {\mathrm{Disp}}_{\psi }\left(q\right)$$*where ${\mathrm{Disp}}_{\psi }\left(q\right):=∥\left(q-q_{\psi }\right)\psi ∥$ is the dispersion of the observable q in the state ψ and $q_{\psi }:=\langle \psi ,p\psi \rangle$ is the ψ–mean values of q and similarly for p.*

**Proof.** One then has
$$|2i\langle \psi ,\partial \omega_{\Lambda }\psi \rangle \left|=2\right|\langle \psi ,\left[p,q\right]\psi \rangle \left|=2\right|\langle \psi ,\left[\left(p-p_{\psi }\right),\left(q-q_{\psi }\right)\right]\psi \rangle |$$
$$\leq 2|\langle \psi ,\left(p-p_{\psi }\right)\left(q-q_{\psi }\right)\psi \rangle \left|+2\right|\langle \psi ,\left(q-q_{\psi }\right)\left(p-p_{\psi }\right)\psi \rangle |$$
$$=2|\langle \left(p-p_{\psi }\right)\psi ,\left(q-q_{\psi }\right)\psi \rangle \left|+2\right|\langle \left(q-q_{\psi }\right)\psi ,\left(p-p_{\psi }\right)\psi \rangle |$$
$$\leq 4∥\left(p-p_{\psi }\right)\psi ∥\;∥\left(q-q_{\psi }\right)\psi ∥$$ □

**Theorem** **6.**
*X is moment-symmetric if and only if $a^{0}=0$.*

*If X is moment-symmetric, then its quantum decomposition is*

$$X=a^{+}+a^{-}$$

*which is the usual expression for the field operators.*


**Remark** **6.**
*As is clear from Equation (Equation 21), momentum comes from the complexification of the real linear map $X:v\in R^{d}\mapsto X_{v}$. When d=1, as in our case, one fixes a linear basis $0\neq e_{1}\in R$ and identifies X with $X_{e_{1}}$. With these notations, $P_{X}:=X_{ie_{1}}$ and is Hermitean for symmetric X. Mathematically, $X_{ie_{1}}$ is a linear operator for non-symmetric X, however, in this case it has no physical (and probably no probabilistic) interest, as one can prove that there exists no complexification of the map $\lambda \in R\mapsto X_{\lambda e_{1}}$, compatible with the relations ${\left(a_{e_{1}}^{+}\right)}^{*}=a_{e_{1}}$, ${\left(a_{e_{1}}^{0}\right)}^{*}=a_{e_{1}}^{0}$ and making $X_{ie_{1}}$ Hermitean.*


### 3.8. The Quantum Mechanics Associated with a Classical Symmetric Real-Valued Random Variable X

In this section, we briefly summarize the correspondence between objects and notions in usual quantum mechanics and the corresponding ones in the theory of orthogonal polynomials.
–*X* classical real-valued random variable (position operator)–$\left(\Phi_{n}\right)$ orthogonal polynomial basis (*n*–particle vectors)–orthogonal polynomial gradation
$$P=\underset{n\in N}{⨁}C\cdot \Phi_{n}\;\left(∼\Gamma (C)\;\mathrm{Boson}\;\mathrm{Fock}\;\mathrm{space}\;\mathrm{in}\;\mathrm{the}\;\mathrm{Gaussian}\;\mathrm{case}\right)$$–CAP operators $a^{+},a,a^{0}$ (creator, annihilator, preservator).–Quantum decomposition of *X*.
(23)$$X=a^{+}+a^{0}+a^{-}$$–generalized Boson commutation relations $\left[a^{-},a^{+}\right]=\partial \omega_{\Lambda }$ and $\left[X,P_{X}\right]=2i\partial \omega_{\Lambda }$.–If *X* is symmetric, one defines the associated conjugate momentum by
$$P_{X}:=i\left(a^{+}-a\right)$$
having them, one can introduce all operators of physical interest, such as $P_{X}^{2}/2$ kinetic energy, V(X) potential energy, Hamiltonians
$$H:=\frac{1}{2}P_{X}^{2}\;;\;H_{harm\;osc}:=a^{+}a$$
and so on. With these, one can study the corresponding generalized Schrödinger equations, although we will not discuss them in this paper.

In other words, every classical random variable canonically determines its own quantum mechanics.

#### 3.8.1. Quantum Correlations and Quantum Covariance of a Classical Random Variable

Let us consider the case of a moment-symmetric classical random variable *X*. Then, by Theorem 6, its quantum decomposition is
$$X=a^{+}+a$$
and if ψ is any unit vector in the Hilbert space, one can compute the classical ψ-variance of *X*
(24)$$\langle \psi ,X^{2}\psi \rangle =\langle \psi ,{\left(a^{+}+a\right)}^{2}\psi \rangle =$$
$$=\langle \psi ,{\left(a^{+}\right)}^{2}\psi \rangle +\langle \psi ,a^{2}\psi \rangle +\langle \psi ,a^{+}a)\psi \rangle +\langle \psi ,aa^{+}\psi \rangle =$$

Thus, we see that the classical ψ-variance of *X* is expressed in terms of what we could call the *quantum ψ-covariance matrix* of the classical random variable *X* with respect to an arbitrary vector state ψ:$$\left(\begin{array}{cc} \langle \psi ,a^{+}a\psi \rangle & \langle \psi ,a^{+}a^{+}\psi \rangle \\ \langle \psi ,aa\psi \rangle & \langle \psi ,aa^{+}\psi \rangle \end{array}\right)$$

Denoting by $\left({\tilde{\Phi }}_{n}\right)$ the normalized sequence of orthogonal polynomials, choosing ψ to be the vacuum vector $\Phi_{0}={\tilde{\Phi }}_{0}$, and using the mutual orthogonality of the vectors $\Phi_{n}:=a^{+n}\Phi_{0}$, the Fock property $a\Phi_{0}\overset{\left(11\right)}{=}0$ reduces the quantum $\Phi_{0}$-covariance matrix of *X* to
$$\left(\begin{array}{cc} 0 & 0\\ 0 & \langle \Phi_{0},aa^{+}\Phi_{0}\rangle \end{array}\right)=\left(\begin{array}{cc} 0 & 0\\ 0 & \omega_{1} \end{array}\right)$$
and one can easily see from (Equation 24) that the $\Phi_{0}$-variance of *X* is:$$\langle \Phi_{0},X^{2}\Phi_{0}\rangle =\langle \Phi_{0},{\left(a^{+}+a\right)}^{2}\Phi_{0}\rangle =$$
$$=\langle \Phi_{0},{\left(a^{+}\right)}^{2}\Phi_{0}\rangle +\langle \Phi_{0},a^{2}\Phi_{0}\rangle +\langle \Phi_{0},a^{+}a)\Phi_{0}\rangle +\langle \Phi_{0},aa^{+}\Phi_{0}\rangle =$$
(25)$$=\langle \Phi_{0},aa^{+}\Phi_{0}\rangle =\omega_{1}$$

Choosing ψ to be a generic normalized *n*-particle vector ${\tilde{\Phi }}_{n}$ (n≥1) and using the multiplication table (Equation 13), (Equation 14), and (Equation 16), we can find the quantum ${\tilde{\Phi }}_{n}$–covariance matrix of *X*:(26)$$\left(\begin{array}{cc} \langle {\tilde{\Phi }}_{n},a^{+}a{\tilde{\Phi }}_{n}\rangle & 0\\ 0 & \langle {\tilde{\Phi }}_{n},aa^{+}{\tilde{\Phi }}_{n}\rangle \end{array}\right)=\left(\begin{array}{cc} \omega_{n}/\omega_{n}! & 0\\ 0 & \omega_{n+1}/\omega_{n}! \end{array}\right)$$
and it is apparent from (Equation 24) that the ${\tilde{\Phi }}_{n}$-variance of *X* for n≥1 is:(27)$$\omega_{n}!\langle {\tilde{\Phi }}_{n},X^{2}{\tilde{\Phi }}_{n}\rangle =\omega_{n}!\langle \Phi_{n},X^{2}\Phi_{n}\rangle =\omega_{n}!\langle \Phi_{n},{\left(a^{+}+a\right)}^{2}\Phi_{n}\rangle =$$
$$=\omega_{n}!\left(\langle \Phi_{n},{\left(a^{+}\right)}^{2}\Phi_{n}\rangle +\langle \Phi_{n},a^{2}\Phi_{n}\rangle +\langle \Phi_{n},a^{+}a)\Phi_{n}\rangle +\langle \Phi_{n},aa^{+}\Phi_{n}\rangle \right)$$
$$=\omega_{n}!\left(\langle \Phi_{n},a^{+}a\Phi_{n}\rangle +\langle \Phi_{n},aa^{+}\Phi_{n}\rangle \right)=\omega_{n}!\langle \Phi_{n},\left\{a,a^{+}\right\}\Phi_{n}\rangle$$
$$\Leftrightarrow \langle {\tilde{\Phi }}_{n},X^{2}{\tilde{\Phi }}_{n}\rangle =\frac{\omega_{n+1}+\omega_{n}}{\omega_{n}!}$$

In usual quantum mechanics $a^{+}a$ is an observable that differs from $aa^{+}$ by an additive constant, and $a^{+}a$ is interpreted as the *number of quanta*, such that its mean value in a quantum state provides the density of quanta in that state, and similarly for the anti-commutator $\left\{a,a^{+}\right\}$. Formulas (Equation 25) and (Equation 27) express the mean values of these quantum observables in terms of numerical characteristics of the classical random variable *X*.

The formula expressing the variance of *X* as a function of quantum correlations has been extended arbitrarily to all classical even moments (recall that for symmetric random variables, the odd moments are zero) in [28]:(28)$$E\left(X^{2n}\right)=\sum_{all\;non-crossing\;pair\;partitions\;{\left(r_{k},l_{k}\right)}_{k=1}^{n}\;of\;\left\{1,\ldots ,2n\right\}}\prod_{k=1}^{n}\omega_{2k-l_{k}}$$Recall that a pair partition ${\left(r_{k},l_{k}\right)}_{k=1}^{n}$ of {1,…,2n} is called non-crossing if, for h≠k,
$$\left[r_{h},l_{h}\right]\cap \left[r_{k},l_{k}\right]\neq \emptyset \Rightarrow \left\{\begin{array}{c} \mathrm{either}\left[r_{h},l_{h}\right]\subset \left[r_{k},l_{k}\right]\\ \mathrm{or}\left[r_{k},l_{k}\right]\subset \left[r_{h},l_{h}\right] \end{array}\right.$$
where [a,b]:={n∈N:a≤n≤b}.

From (Equation 26), we can recognize that the right hand side of (Equation 28) is *a sum of products of quantum pair correlations* of the form $\omega_{n}!\langle {\tilde{\Phi }}_{n},a^{+}a{\tilde{\Phi }}_{n}\rangle =\langle \Phi_{n},a^{+}a\Phi_{n}\rangle$.

Recall that the property that each 2n-moment of a symmetric classical random variable is expressed as sums, over all pair partitions of {1,…,2n}, of products of *classical pair correlations*, which characterizes classical mean-zero Gaussian states.

In this sense, Formula (Equation 28) is a generalized form of Gaussianity in the sense that, with respect to usual Gaussianity, one replaces:–the pair partitions of {1,…,2n} by the non–crossing ones;–the classical pair correlations by quantum pair correlations;–the pair correlations with respect to a single state by the pair correlations with respect to a sequence of states.

In this sense, in [29], the authors spoke of *Gaussianization of probability measures*.

This generalized form of Gaussianity has not been studied in the literature, even in the simplest case when the sequence $\left(\langle {\tilde{\Phi }}_{n},\;\cdot \;{\tilde{\Phi }}_{n}\rangle \right)$ is replaced by a single pair of states $\langle \psi_{1},\;\cdot \;\psi_{1}\rangle$, $\langle \psi_{2},\;\cdot \;\psi_{2}\rangle$.

## 4. Fermions

In [30] it is proven that, while for a single random variable (system with 1 degree of freedom, the only ones considered in the present paper) the Fermi commutation relations (CAR) can be deduced from the canonical quantum decomposition of a classical Bernoulli random variable (in fact, for the validity of this decomposition, it is not required that all the $\omega_{n}$ are ≠0), this result cannot be true for a vector valued random variable (random field) taking values in a space of dimensions strictly greater than 1 (many degrees of freedom). In that case, the CAR become
$$\left\{a_{j}^{+},a_{k}\right\}=\delta_{j,k}\;,\;j,k\in D:=\;\mathrm{finite}\;\mathrm{set}\;$$

In the same paper, it is proven that the multi-dimensional CAR follow from a simple and meaningful (physically and probabilistically) algebraic condition, namely, a weak form of the Pauli exclusion principle.

This fact raises new questions for classical probability and quantum physics. In fact, the Pauli exclusion principle can be interpreted as an algebraic constraint on the family of random variables of a classical Bernoulli process.

This introduces a new idea from quantum mechanics into classical probability. In fact, one of the basic principles of classical probability is that *all constraints among random variables of a process X (interactions) are included in their probability distribution*. A typical example is a two-component vector random variable *X* on the unit circle; in Kolmogorov representation, the state space of *X* is R×R (no kinematical constraints), with the support of its probability distribution concentrated on the unit circle. More generally, while in classical probability every process can be described by a product space, the probability measure on the space may not be of the product’s type. The algebraic analogue of this statement would be that while any quantum stochastic process can be described on a tensor product space, the state on it need not be of tensor product type. The above-mentioned result shows that in quantum probability the situation is different. A non-trivial consequence of this remark is the possibility of a physical interpretation of the various forms of algebraic constraints appearing in various notions of stochastic independence in quantum probability (i.e., boolean, free, monotone). In [30], this idea is illustrated with concrete examples.

## 5. Feedback for Physics

In the previous sections, it has been shown that:–usual Boson QM coincides with the probabilistic quantization associated with the Gauss–Poisson class;–usual Fermion QM coincides with the probabilistic quantization associated with the Bernoulli class (plus the weak form of the Pauli principle).

Let us consider theGgaussian family. This is characterized as the unique symmetric measure with the principal Jacobi sequence $\omega_{n}\overset{\left(19\right)}{=}n\hbar ,\;\forall n\in N$, defining the Heisenberg commutation relations (Theorem (4)). Because the commutation relations are determined not by the sequence $\left(\omega_{n}\right)$, but by $\left(\partial \omega_{n}\right)$ (see (Equation 17)), adding a constant does not change the commutation relations. Thus, one can define the *Gaussian class* as the class of probability measures on the real line such that their principal Jacobi sequence is a first order polynomial in *n*:$$\omega_{n}=a_{1}n+a_{0}\geq 0\;,\;\forall n\geq 1\;,\;\omega_{0}:=0$$A very natural generalization of this class consists of those probability measures on R such that their principal Jacobi sequence is a polynomial in *n* of arbitrary degree d∈N.

Within this class, the case d=2 is critical because it is the highest value of *d* such that the ∗–Lie algebra generated by *a* and $a^{+}$ is finite dimensional. In fact, in this case this ∗–Lie algebra coincides with a (necessarily trivial) central extension of sl(2,R) (see [31]). The associated quantum mechanics have been developed in a series of papers by several authors (see [32] and references therein). The classical random variables in this class are exactly the three non-standard Meixner families of Meixner–Pollaczek probability measures (Gamma, negative binomial (or Pascal), and Meixner), which have been widely studied in classical probability theory.

## 6. Discrete Observables Canonically Associated with Continuous Classical Random Variables

From the multiplication table for *a* and $a^{+}$, it follows that the observable (Hermitean operator)
$$aa^{+}=\omega_{\Lambda +1}$$
has a pure point spectrum provided by
(29)$$\omega_{\Lambda }=\left\{\omega_{n}:n\in N\right\}$$
and the corresponding eigenvectors are the orthogonal polynomials. However, *X* can be a continuous random variable (for example, the uniform distribution of the interval [0,1]). Thus, when considered as a multiplication operator, it has a continuous spectrum.

This shows that all continuous real valued random variables with all moments are canonically associated with random variables which are functions of the number operator Λ, and hence, with a pure point spectrum. In the boson case, $\Lambda =a^{+}a$ up to a positive multiple, and this observable is interpreted as the number of quanta.

Because these observables appear in probabilistic quantization, the following problem naturally arises: what are the quanta in a specific classical (say, economical or psychological) framework?

In boson quantum mechanics, the presence of both discrete and continuous observables is the counterpart of the wave–particle duality in the mathematical formalism. This suggests the speculation that such a duality may have a meaning outside quantum mechanics.

## 7. Conclusions

For about 100 years, we have been accustomed to seeing quantization as something specific to microscopic physics. Now, we know that this is not true. Wherever there is probability, there is quantization. In this sense, we can speak of the ubiquity of probabilistic quantization.

### Open Challenges

Physicists have learned over almost 100 years to understand the physical meaning of quantum correlations (at least to a certain extent). Now that we know that every classical random variable has its own quantum correlations (see the discussion in Section 3.8.1 for examples of quantum correlations associated with classical random variables), we have to learn how to interpret them in the different fields of application of classical probability theory, from information theory to economics, from machine learning to biology, etc.

Even if the main message of this paper is that every classical random variable determines its own quantum mechanics, it should be noted that the deeper quantum structure of classical probability allows for new results in classical probability to be obtained using quantum methods. Already, there exist several examples of open problems in classical probability that have been solved using quantum methods, and it is natural to expect that in the future these kind of results will proliferate.

## Data Availability

Already covered by bibliography.

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
