# Peer review of "New Challenges for Classical and Quantum Probability"

_entropy, 2022, doi:10.3390/e24101502_

Round 1

Reviewer 1 Report

Summary:

This paper presents an interesting approach to the classical probability theory based on quantum probability. In Sec. 2 and 3, an introduction as well as author's contributions to the subject are given. The basic mathematical background is shortly explained from Sec. 3.1 to 3.3. Quantum interpretation of the Jacobi monic relation is pointed out in Sec. 3.4, and it then explains a commutation-like relation for a classical random variable in Sec. 3.5. In particular, the appearance of the (boson) Fock space from the Jacobi monic relation is very clearly explained. In Sec. 3.6, The standard Heisenberg-like relation is analyzed under the assumption of a positive constant. The conjugate operator is also shown to exist in the classical probability theory for a symmetric real valued random variable in Sec. 3.7. This then proves an uncertainty relation for the classical random variable. Sec. 3.8 and 3.9 seem to give a brief outlook of the proposed method. Sec. 4, Sec. 5, and Sec. 6 list several topics related to the subject without detail accounts. 

Over all impression and recommendation:

Generally speaking, I enjoy reading this paper. In particular, I appreciate author's efforts on understanding quantum theory from the classical probability theory over the past years. This manuscript might be a useful resource for young researchers who are not familiar with the subject. Before I recommend it to be accepted, I do have several comments to be incorporated. In particular, I kindly ask the author to improve some parts of the manuscript, since I am unable to understand clearly. The organization of the paper should be improved so that readers can benefit your paper. Please see the my comments in detail below. 

Major comments:

1) Structure: 

The structure of the paper seems to be a bit strange at this stage. Sec. 1 and 2 are fine. But then, Sec. 3, 4, 5 and 6 need to be improved. For example, Sec. 3.8 is too short and this section seems to me just an outlook of the paper. Furthermore, Sec. 3.9 is the conclusion section. Title of Sec. 3, ``Some notations'' might be revised by a more appropriate one, since this section present the main result of the paper. Then, the author start sections 4, 5, and 6 after the conclusion. I find that Sec. 4, 5, and 6 are also too short to be independent sections. They can be subsections of the outlook of the paper. If possible, I suggest the author to focus on more concrete outlook rather than a list of the very general and diverse topics. I understand your approach is very general, but readers won't understand Sec. 4 and 5. 

2) Use of bold and italic fonts:

This is my personal opinion, but I find it hard to read the current writing with so many bold and italic fonts. In particular, I am not able to understand the use of bold fonts. Please revise the manuscript so that only necessary words use bold and italic fonts. Besides this writing format matter, I also find that there are so many unnecessary line breaks, which make me hard to read the manuscript. 

3) The role of complex numbers:

I appreciate if the author clarifies the role of complex numbers in your approach. Do you only consider a real-valued random variable and a real-valued measure?  Above Eq. (3.1), the map takes values in the complex number. This is because the author introduce complex valued polynomials of the random variable without any reasoning. I think this assumption together with the introduction of an inner product defines a complex Hilbert space structure. Thereby, we anticipate the desired result already. 

4) New findings:

I understand that the present paper gives also a summary of the author's contribution to the subject. However, it is necessary to clearly state what is a new finding of the current paper as a research paper. If there is nothing new added at this stage, one should publish it as a review. It seems to me that the basic idea for the use of the Jacobi monic relation was already presented. If so, please cite necessary references more appropriately. 

5) References:

I understand that the subject of the present paper is largely done by the author and his collaborators. Yet, citation of own 17 papers out of 31 total references is not usual. Please select only necessary own papers. I find that some of the cited papers are not accessible, since they are not published as journal papers. Please also note that the subject of constructing more general probabilistic theory including quantum mechanics has been actively studied. Please also add necessary references on the related topic(s) if necessary. 

Other comments:\\

- Title: The submitted title is ``New challenges for classical and quantum probability Talk given at the Conference: Quantum Information and Probability: from Foundations to Engineering''. Please check if this is correct. 

- Eq. (22) on page 18: The middle integration interval should be ``R'' without ``)''. 

- Line 350 on page 19: $\hbar$ is Planck's constant divided by $2\pi$ to be precise. (Minor)

- Line 374 on page 20: ``re construction'' should be a single word. 

- Sec. 3.2: Orthogonal polynomials satisfying the three term recursion relation contains the Hermite polynomial as a special case ($a^0$ case). Is this correct? If so, could you add a remark on it. 

- Line 415 on page 23: Do you need to cite a reference for Favard's result? 

- Eq. for ``preservation operator'': the right hand side should be $\alpha_n \Phi_n(x)$. 

- Line 429 on page 25: ``The identity (3.9)'' should be ``The identity (3.7)''.

- Line 435 on page 25: Delete ``aaa''. 

- At many places, the operators are required to be Hermitian rather than self-adjoint. Why is this so? 

- Theorem 4: It is interesting to see the appearance of the usual CCR, if one adopts a sequence of numbers with common difference. What about other choices: can you obtain other systems? 

- Please clarify the reason why one uses the symmetric classical random variable only. For example, the discussion in Sec. 3.7 only state the result for this particular case without any explanation. To ask differently, can you find a momentum operator for non-symmetric case? 

- Eq. (3.18) on page 30: Please define the symbol $\psi$. The right hand side should be square root? 

- Sec. 3.8: The author intends to present the correspondence in the proposed theory, but it is rather too short to understand. If this is already explored in details, please give a necessary reference(s), or just put it as an outlook of the paper. 

- Sec. 3.9.1: This subsection should be the last part of the paper as it conveys the massage of the author. 

- Sec. 3.9.2: This subsection seems to be incomplete. What is the main conclusion out of this calculation? 

- Sec. 4: Please select what to be written with a concrete massage. The current style of section 4 is not understandable. The conjecture here is also unspecific. (I suggest it to be included as a list of outlooks.)

- Sec. 5: Please revise it or remove it. What do you mean by ``higher powers''? Do you mean the more general particle statistics like para-statistics in quantum field theory? 

- Sec. 6: This section is rather unsatisfactory. There are huge list of literature on wave-particle duality. At this stage, it is not appropriate to attribute the result of this paper to wave-particle duality. The claimed relation is rather superficial at this stage. 

Reviewer 2 Report

This is a nice review paper, reporting in particular new and interesting idea linkong the quantisation with classical real variables. The idea is based on operator interpretation of Jacobi identity concerning orthogonal polynomials of this variable.  Jacobi identity gives canonical quantum decomposition of classical random variable in terms of creation, anihilation and preservation operators. In this way one comes to the conclusion that non - commutativity and quantumness arises naturally from clasical probability.

Unfortunately, the manuscript is is well prepared. It looks like a draft not a final verion of the paper. Therefore I think that the author should more carefully prepare the manuscript and then the paper can be accepted fro publication.

Reviewer 3 Report

For the first look, the style of this work is not like a regular paper.

1) The structure of the paper is bad. Please check the title page. The talk given at the conference should be written in a standard writing style not the present form.

2) After reading this paper, the authors should find that the PDF is bad, each page has only almost one half of content. 

I suggest the author revises this and then resubmit. 

Round 2

Reviewer 1 Report

I would like to thank the author to revise the manuscript. I have read your reply and the manuscript. This revised version is much more improved compared with the first version and can be published after minor revision. There are still a few places that need revision. Please go through the whole manuscript to check the presentation. 

For example, the title still sounds odd to me. The author should also select a review paper rather than an article. References need to be checked again as some of them are not in the correct format. 

Author Response

Corrections included.

-- Erased:   \\
Talk given at  the Conference:Quantum Information and Probability:from Foundations to Engineering

--  Inserted:\\
\keyword{keyword 1; keyword 2; keyword 3 (quantum probability, orthogonal polynomials,
quantum decomposition of a classical random variable
%List three to ten pertinent keywords specific to the article; yet reasonably common within the %subject discipline.
)}

-- I have corrected all the mis-spellings I was able to find with Spell Check.

-- I have simplified the proof of Theorem 3 and shortened the Remark before it.

-- I have deleted the following references:\\

\bibitem[Ac81a-Torino]{Ac81a-Torino}
Accardi L.:\\
Foundations of quantum probability,\\
Rendiconti del Seminario Matematico  dell'Universit\`a  e del Politecnico, Torino \\
special issue dedicated to the Conference: Stochastic Problems in Mechanics, Torino 28-30 Maggio
1981, (fasc. e) (1982) 249--270\\

\bibitem[Ac93-Lecce]{Ac93-Lecce}
Accardi L.:
On the axioms of probability,
in: The Foundations of Quantum Mechanics, Proceedings of
the Lecce Conference:
I fondamenti della meccanica quantistica: analisi storica e programmi aperti (1993),
C. Garola, A. Rossi (eds.) Kluwer (1995) 1-18\\
Volterra Preprint N. 194 November (1994)

\bibitem[Ac95a-Roma-Fund-Prob-QM]{Ac95a-Roma-Fund-Prob-QM}
Accardi L.:  \\
Can mathematics help solving the interpretational problems of quantum theory?\\
Il Nuovo Cimento B (11) 110 (5-6) (1995) 685-721 \\

\bibitem[Ac04b-Axioms for QP]{Ac04b-Axioms for QP}
Luigi Accardi:\\
Axioms for quantum probability,\\
in: Proceedings of the  "25th QP CONFERENCE "QUANTUM PROBABILITY AND RELATED TOPICS",\\
June, 20-26, 2004, Bedlewo, Poland,\\
Banach Center Publications 73 (2006) 15-33

-- I have corrected many misprints and introduced several minor linguistic corrections in
the exposition.

Reviewer 3 Report

I suggest the authors submit it to philosophical journals. It is found that the author did not know the professional writing. It is the first time to see the title in "...Talk given at the ...........". The keywords should be provided explicitly. 

Author Response

(The authors gave the same response as above.)
